# Experiment and Molecular Dynamic Simulation on Interactions between 3,4-Bis(3-nitrofurazan-4-yl) Furoxan (DNTF) and Some Low-Melting-Point Explosives

**DOI:** 10.3390/molecules29163757

**Published:** 2024-08-08

**Authors:** Junming Yuan, Runsheng Huang, Jinying Wang, Xiwei Xing, Jing Wang, Tao Han, Qi Yang, Jia Yang

**Affiliations:** School of Environmental and Safety Engineering, North University of China, Taiyuan 030051, China; 17835131200@163.com (R.H.); wjywzhy@126.com (J.W.); 15903412016@163.com (X.X.); wj1872691402@163.com (J.W.); h3581305@163.com (T.H.); m17636191448@163.com (Q.Y.); 16603596671@163.com (J.Y.)

**Keywords:** low-melting-point explosives, molecular dynamic simulation, interaction, compatibility, mechanical sensitivity

## Abstract

3,4-bis(3-nitrofurazan-4-yl) furoxan (DNTF) is an explosive with excellent performance, and the use of DNTF as a high-energy component is of great significance for improving the comprehensive performance of weapons. To explore the effect of DNTF on low-melting-point molten carrier explosives, the compatibility between DNTF and other low-melting-point explosives was analyzed by differential scanning calorimetry, and mechanical sensitivity was tested. The compatibility and cohesive energy density between DNTF and other low-melting-point explosives were calculated by Materials Studio. The results showed that DNTF has good compatibility with most low-melting-point explosives, and the peak temperature change of the mixed system formed by melt-casting is not obvious. Among them, DNTF has the best compatibility with MTNP, TNT, and DNAN; moderate compatibility with DFTNAN and DNP; and the worst compatibility with DNMT. The sensitivity test results indicate that the combination of DNTF and TNT has the most significant reduction in mechanical sensitivity. DFTNAN and MTNP have better stability than DNTF and can generate strong interaction forces with DNTF. Other low-melting-point explosives mixed with DNTF have lower intermolecular forces than DNTF. The DNTF/MTNP system requires the most energy to phase change when heated compared to other mixed systems and is the least sensitive to heat. The DNTF/DNMT system has the lowest cohesive energy density and is the most sensitive to heat.

## 1. Introduction

3,4-bis(3-nitrofurazan-4-yl) furoxan (DNTF) is a new type of high-energy-density material [1,2,3], which has a stronger working ability compared to octahydro-1,3,5,7-tetranitro- 1,3,5,7-tetrazocine (HMX) [4] and has a significant accelerating effect on metals. DNTF benefits from high detonation velocity and energy, high density, good oxygen balance, a low melting point, and moderate sensitivity. It can be used as the main component of cast explosives [5,6,7,8] and high-energy propellants [9,10,11] and has broad application prospects [12,13]. Therefore, DNTF has received widespread attention and research.

DNTF is a hydrogen-free explosive with a low melting point, moderate sensitivity, good thermal stability, and indirect steam heating that does not decompose for a long time. It is suitable for melting and casting explosive formulations. DNTF can be mixed with 2,4,6-trinitrotoluene (TNT) explosives to form low-cocrystal compounds that can be formulated into liquid carriers with different melting points and energy requirements [14]. By replacing TNT with DNTF as a liquid phase carrier, the energy of mixed explosives will rise to a new level, which is of great significance for improving the overall performance of weapons [15]. As a low-melting-point and high-energy explosive, DNTF meets the requirements of the casting process itself, but its high sensitivity greatly increases the hazard of the explosive. The chemical and crystal structures of DNTF explosive molecules are shown in Figure 1 [16].

The use of DNTF as a high-energy component has the following advantages: (1) It has better solid-phase uniformity compared to the addition of solid particles to the melt-cast carrier at lower temperatures. High-sensitivity explosives and low-sensitivity explosives can be melted and cast to improve their sensitivity; (2) Compared with the addition of high-energy solid particles to the melt-cast carrier, DNTF and the melt-cast carrier together melt-cast to form a lower hardness and better ductility, which means it has better mechanical properties; (3) DNTF has excellent detonation performance and higher energy than hexahydro-1,3,5-trinitro-1,3,5-triazine (RDX).

According to current research on the compatibility of DNTF, it has been found that DNTF/TNT and DNTF/5-ATEZN (5-Amino-tetrazolium Nitrate) have good compatibility, DNTF/NTO (3-Nitro-1,2,4-triazol-5-one) and DNTF/TATB (2,4,6-Triamino-1,3,5-trinitrobenzene) have moderate compatibility, and DNTF/LLM-105 (2,6-Diamino-3,5-Dinitropyrazine-1-Oxide) and DNTF/PVAC (Polyvinyl Acetate) have poor compatibility [17,18,19]. The mechanical and thermal sensitivity of DNTF-based binary cocrystal materials decreases with increasing DNTF content. The maximum bond length of the N-O bond initiated by DNTF can serve as a theoretical criterion for the relative mechanical sensitivity and thermal safety of DNAN/DNTF binary cocrystal materials [20].

Existing research has shown that the addition of other substances has a significant impact on the performance of DNTF [21]. For example, molecular dynamics simulation methods are used to establish molecular models of NC (Nitrocotton)/DNTF [22] and GAP (Glycidyl azide polymer)/DNTF [16] blends to explore the interaction between additives and explosives.

The above research results are related to the performance of DNTF explosives and their compatibility with other explosives. Similar research methods have also been found in the literature [23,24,25,26], which respectively explore the compatibility issues of 2,3-bis (hydroxymethyl)-2,3-dinitro-1,4-butanediol tetranitrate (DNTN), dihydroxylamonium 5,5′-bistetrazole-1,1′-diolate (TKX-50), tetraethylammonium decahydrocarbonate (BHN), and trans-1,4,5,8-tetranitro1,4,5,8-tetrazadecalin (TNAD) with other energetic components and inert materials. These studies have important reference significance for the research of this paper. In addition, the ab initio molecular dynamic method can also be used to study the compatibility between components of energetic materials [27,28].

In this article, a mixture of DNTF and six types of low-melting-point explosives was prepared. The compatibility and interaction between DNTF and other low-melting-point explosives was investigated using electron microscopy morphology characterization and DSC analysis. Combined with experimental characterization and molecular dynamics simulation calculations, the physical and chemical interactions between DNTF and low-melting-point explosives were studied at different temperature ranges. The research results can provide certain theoretical support for the safe use of DNTF.

## 2. Experimental Methods

### 2.1. Materials and Equipment

Materials: 3,4-bis(3-nitrofurazan-4-yl) furozan (DNTF), 1-methyl-3,4,5-trinitropyrazole (MTNP), 1-methyl-3,5-dinitro-1,2,4-triazole (DNMT), 2,4,6-trinitrotoluene (TNT), 2,4-dinitroanisole (DNAN), and 3,4-dinitropyrazole (DNP) samples were provided by Xi’an Modern Chemistry Research Institute and purified by recrystallization in alcohol whose mass fraction purity was greater than 0.995. 3,5-difluoro-2,4,6-trinitroanisole (DFTNAN) was produced by the North University of China. The mixtures of DNTF and some low-melting-point explosives were prepared according to a mass ratio of 1:1.

Equipment: EM-30 scanning electron microscope from COXEM Company in Daejeon, South Korea; HCT-1 differential scanning calorimeter from Beijing Hengjiu Technology Co., Ltd. (Beijing, China).

### 2.2. Performance Characterization Methods

SEM: The particle morphology of DNTF and low-melting-point-explosive samples was observed by scanning electron microscopy (SEM) and compared with the morphology of DNTF raw materials. The samples were treated with gold spray and accelerated to a voltage of 15 kV.

Mechanical sensitivity: In accordance with methods 601.2 and 602.1 of GJ772A-97, the impact sensitivity and friction sensitivity of DNTF and DNTF with low-melting-point-explosive samples were tested using an MYG-I impact sensitivity tester and an MYG-I friction sensitivity tester, respectively. The characteristic drop height test used a 5.000 ± 0.005 kg drop hammer with a dosage of 30 ± 1 mg; the friction sensitivity test pressure was 3.92 MPa, the swing angle was 90°, and the dosage was 20 mg.

DSC: The thermal performance parameters and thermal decomposition process of the samples were tested and analyzed using DSC HQC-1. DNTF, DNMT, TNT, DNAN, DNP, MTNP, and DNTF mixed with low-melting-point explosives were tested and analyzed using DSC at a heating rate of 10 °C/min. The temperature range was 30–350 °C, with a dosage of 3 ± 0.1 mg. The experiment used a sealed alumina crucible and an empty crucible as reference material.

### 2.3. Evaluated Standard of Compatibility for Mixtures

The evaluated standards [23] of compatibility are listed in Table 1.

The temperature change of the decomposition peak is calculated as:(1)$$∆T_{p}=\left|T_{p1}-T_{p2}\right|$$
where ${∆T}_{p}$ is the change in decomposition peak temperature of the system alone relative to the mixed system, K; $T_{p1}$ is the decomposition peak temperature of the separate system, K; and $T_{p2}$ is the decomposition peak temperature of the mixed system, K.

## 3. Molecular Dynamic Simulation

### 3.1. DNTF/Low-Melting-Point Explosives Mixed Models

The molecular structural formulas and configurations of six low-melting-point explosives provided by the laboratory—DFTNAN, TNT, DNMT, DNAN, MTNP, and DNP—are shown in Figure 2.

The amorphous cell is used to create an amorphous model of DNTF and low-melting-point explosives, ensuring that the density of the model is close to the theoretical density. The initial model established is shown in Figure 3.

The established initial model parameters are shown in Table 2.

### 3.2. Molecular Dynamics Calculations

The interaction between DNTF and low-melting-point-explosive components at low temperatures was studied by molecular dynamics simulation using Materials Studio2019 software. The COMPASS force field and a smart minimization method were used to optimize the geometry of the amorphous model with a convergence limit of 0.001 kcal/mol. The MD simulation used the Nosé–Hoover temperature control method and the Berendsen pressure control method. The van der Waals and electrostatic interactions were calculated using the atom-based and Ewald methods, respectively. The cutoff radius was set to 12.5 Å, the spline width was set to 0.1 nm, the buffer width was set to 0.05 nm, the step size was set to 1 fs, and the isothermal isobaric (NPT) ensemble was selected. The temperature was 298.15 K, and the pressure was 101 KPa. The total simulation time was 300 ps. In order to eliminate the influence of unreasonable energy and volume on the calculation results, a balanced system of 100 ps was selected to calculate the relevant performance.

## 4. Results and Discussion

### 4.1. Electron Microscopy Morphology

SEM was used to characterize the microstructure of DNTF explosive particles mixed with low-melting-point-explosive particles. The electron microscope photos of the morphology characteristics of DNTF explosive grains are shown in Figure 4a,b, which are photos of DNTF grains magnified by 200 and 500 times, respectively. Overall, the DNTF crystal exhibits a layered morphology, and the surface of the uncoated DNTF is relatively smooth and flat. The particles have a regular polyhedral shape with slightly smaller grains attached to the surface without any other attachments.

The electron microscopy morphology characterization of DNTF and low-melting-point explosives is shown in Figure 5. The surface of DNTF/MTNP is the smoothest, with very few small particles, indicating the best compatibility between DNTF and MTNP. After mixing DNTF with TNT and DNMT, the surface roughness increases, and crystalline particles appear on the surface, with the DNTF/DNMT phenomenon being more obvious. The surface roughness of DNTF, DNAN, and DNP is more pronounced, and the surface is clustered. Compared to DNTF/DNAN, DNTF/DNP has smoother surfaces and less clustering. The surface of mixed DNTF and DFTNAN is the roughest, and the crystal surface adheres to incompletely melted explosive particles, indicating the worst compatibility between the two.

### 4.2. DSC Analysis

The thermal degradation of explosives is a complex process that typically involves multiple mechanisms, triggered by similar activation barriers and occurring at close rates. Only one reaction will initiate decomposition and continue for a considerable period of time, especially in the case of crystals. One mechanism usually dominates the other reaction, and even when the two reactions start slightly later, they can switch at higher temperatures to become the main reaction. The differential thermal curves between DNTF and low-melting-point explosives are shown in Figure 6.

According to the DSC curve in Figure 6, the thermal decomposition process of DNTF and low-melting-point explosives is mainly divided into two stages: an endothermic melting stage and an exothermic decomposition stage. The DSC curve of DNTF/DFTNAN shows a single exothermic peak, and the initial decomposition temperature and thermal decomposition peak temperature are significantly higher than those of DNTF. After mixing DFTNAN and DNTF, DNTF and DFTNAN decompose almost simultaneously, and there is an exothermic peak in the DSC curve of the mixture. In the DNTF/DFTNAN mixed system, the peak decomposition temperature of DNTF/DFTNAN was 2.39 °C higher than that of the DNTF alone. According to the compatibility criteria, the mixed system of DNTF and DFTNAN can be considered compatible, and the compatibility evaluation is B. Similarly, the compatibility between DNTF explosives and other low-melting-point explosives was determined according to the ABCD compatibility criterion, and the discrimination results are listed in Table 3.

After mixing TNT, MTNP, and DNTF, the exothermic decomposition peak temperature of the mixed system did not cause a significant shift compared to pure DNTF, and the temperature difference of the exothermic decomposition peak was less than 2 °C. The exothermic decomposition peak temperature of DNTF/DNAN and DNTF/DFTNAN mixed systems was slightly shifted compared to pure DNTF, while the exothermic decomposition peak temperature of DNTF/DNMT and DNTF/DNP mixed systems was significantly shifted compared to pure DNTF. Based on the above compatibility criteria, DNTF has good compatibility with most low-melting-point explosives, and the thermal decomposition peak temperature shift of the mixed system formed by melting and casting is relatively small. Based on the electron microscopy characterization results, DNTF has the best compatibility with MTNP, good compatibility with TNT, moderate compatibility with DNP and DNAN, and a slight deviation in morphology characterization and compatibility results between DNTF, DFTNAN, and DNMT.

### 4.3. Mechanical Sensitivity

The impact friction sensitivity test results of DNTF, low-melting-point explosives, and the mixed system of DNTF and low-melting-point explosives are listed in Table 4.

After mixing DNTF and low-melting-point explosives, the impact sensitivity of the mixed system is arranged in the following order: H_50_(DNTF/DNAN) > H_50_ (DNTF/TNT) > H_50_(DNTF/MTNP) > H_50_(DNTF/DFTNAN) > H_50_(DNTF/DNMT) > H_50_ (DNTF/DNMT) > H_50_ (DNTF/DNP). The order of the friction sensitivity of the DNTF/low-melting-point explosive mixed system is as follows: P (DNTF + TNT) > P (DNTF + DNAN) > P (DNTF/MTNP) > P (DNTF + DNP) > P (DNTF + DNMT) > P (DNTF + DFTNAN).

Combining the sensitivity test results of pure DNTF and pure low-melting-point explosives, the addition of DFTNAN, DNMT, MTNP, and DNP all improved the impact sensitivity of DNTF, while TNT and DNAN slightly decreased the impact sensitivity of DNTF. The friction sensitivity data show that the friction sensitivity of the DNTF and low-melting-point explosive mixture system is reduced compared to pure DNTF, with the DNTF/TNT mixture system showing the most significant reduction in friction sensitivity. The results of the mechanical sensitivity data indicate that the addition of TNT and DNAN can effectively reduce the mechanical sensitivity of DNTF. For explosives such as DFTNAN, TNT, DNAN, and MTNP that have good compatibility with DNTF, the addition of low-melting-point explosives significantly reduces the friction sensitivity of DNTF explosives. The impact sensitivity of DNTF/TNT and DNTF/DNAN mixed systems is lower than that of pure DNTF, while the impact sensitivity of DNTF/DFTNAN and DNTF/MTNP mixed systems is improved.

Combining the results of DNTF and low-melting-point explosive mechanical sensitivity, it can be found that the DNTF/TNT composite system has the most significant effect on reducing mechanical sensitivity. After the addition of TNT, the friction sensitivity of DNTF decreased by 76%, the drop height increased by 3.02 cm, and the impact sensitivity decreased. The impact sensitivity of pure DNTF explosives is not high, like HMX (H_50_ = 43.56 cm), but the high friction sensitivity of DNTF makes it susceptible to friction ignition and explosion, leading to detonation. The addition of TNT significantly reduces the friction sensitivity of DNTF, which helps improve the safety of DNTF explosives.

### 4.4. Binding Energy Calculation

Binding energy ($E_{bind}$) is defined as the amount of work required to pull each component of the system from “zero distance” to “infinity”. $E_{bind}$ is an important parameter to measure the size of the interaction force between different components in the composite system, and it can also predict the compatibility between different components. The higher the $E_{bind}$ value, the more precise the combination between components in the composite system, the stronger the interaction force, the higher the thermodynamic stability of the system, and the more stable the formed structure. The mathematical expression is given in (2):(2)$$E_{bind}=-E_{inter}=-\left(E_{total}-E_{DNTF}-E_{LMPE}\right)$$
where $E_{inter}$ is the interaction energy between the two components; $E_{total}$ is the total energy of the system at equilibrium; and $E_{DNTF}$ and $E_{LMPE}$ are the energies of the remaining parts in the composite system after removing the respective components of DNTF and Layer 1, respectively.

Based on the MD simulation of the substituted models, the corrected binding energies Eb* are summarized in Table 5. The binding energy Eb* is a measure of the interaction strength of the components [29]. It might affect the structural stability of the cocrystal—the larger the Eb* value, the more stable the cocrystal becomes—and provide a general evaluation for screening the preferred substitution pattern and molecular ratio.

The compatibility of the composite system was calculated by molecular dynamics simulation. According to the numerical results of the modified binding energy, it was found that the law is as follows: Eb * (DNTF/MTNP) > Eb * (DNTF/TNT) ≈ Eb * (DNTF/DNAN) > Eb * (DNTF/DFTNAN) > Eb * (DNTF/DNMT) > Eb * (DNTF/DNMT). The binding energy of DNTF with MTNP, TNT, and DNAN composite systems is relatively high, while the binding energy with DNMT is the lowest. Based on the compatibility between DNTF and the low-melting-point explosive composite system in Table 3, MTNP, TNT, DNAN, and DNTF have good compatibility. The binding energy of the three systems is higher than that of other low-melting-point-explosive systems, and the calculation law of binding energy is consistent with the DSC compatibility discrimination results.

At the same time, the binding energy of the composite system at 2000 K and 3000 K is calculated by MS, and the binding energy results with respect to 300 K temperature are presented in Table 6, Table 7 and Table 8.

The calculation results indicate that the binding energy between DNTF and DFTNAN, TNT, DNAN, and DNP systems is higher at a temperature of 300 K, while the binding energy between DNTF and DNMT systems is the smallest, which is consistent with the compatibility discrimination results. Under high-temperature conditions of 2000 and 3000 K, the interaction between DNTF and low-melting-point explosives is affected. A typical representative is TNT and DNAN, which have good compatibility with DNTF. The binding energy of the TNT system decreases with increasing temperature compared to other low-melting-point explosives, while the binding energy of the DNTF/DNAN composite system is higher than that of other low-melting-point-explosive systems. As for the DNMT with the worst compatibility, the binding energy of the DNTF/DNMT system at 2000 K is higher than that of the DFTNAN system with better compatibility, and the binding energy of the DNMT system at 3000 K is higher than that of the TNT system with better compatibility.

### 4.5. Cohesive Energy Density

Cohesive energy density (CED) is a parameter used to measure intermolecular forces, which is the energy required per unit volume of 1 mole of a condensate to overcome intermolecular forces and become gaseous [30]. It characterizes the energy required for condensed state gasification, which in MD simulation is the sum of van der Waals energy and electrostatic energy, i.e., the non-bonding energy between molecules. The lower the CED value of the explosive, the less energy is required for the explosive to undergo a phase transition when heated, which indirectly indicates that the explosive is more sensitive to heat under the same conditions. Based on the MD simulation calculation of the cohesive energy density, the results are shown in Table 9.

According to the calculation results in Table 9, the cohesive energy density is as follows: DNTF/MTNP > DNTF/DFTNAN > DNTF/TNT > DNTF/DNP > DNTF/DNAN > DNTF/DNMT. The cohesive energy density of the MTNP/DNTF composite system is the highest, indicating that the combination of MTNP and DNTF requires the highest energy to undergo phase transition when heated compared to other mixed systems and is the least sensitive to heat under the same conditions. However, the DNTF/DNMT system requires the least energy to undergo a phase transition when heated and is the most heat sensitive under the same conditions. The cohesive energy density of the TNT composite system, which is consistent with the compatibility discrimination results of MTNP, is lower than that of the DFTNAN hybrid system with moderate compatibility. The compatibility discrimination results of DNAN and DFTNAN are consistent, while the cohesive energy density value of the DNTF/DNAN system is lower than that of DNP with poor compatibility and is close to the CED value of the DNMT system, with the worst compatibility.

## 5. Conclusions

By analyzing the comprehensive performance of DNTF explosive, it was confirmed that its mechanical and thermal sensitivity are relatively high, which affects its promotion and use in the ammunition field. In response to this issue, some low-melting-point explosives and DNTF explosives were selected for mixed preparation in this paper, and their morphology, thermal performance, and mechanical sensitivity were characterized and tested. In addition, molecular dynamics was used to simulate and analyze the binding energy of the mixture. Based on the above research work, some conclusions have been drawn, as follows:(1)Combining electron microscopic characterization and DSC analysis, it can be concluded that DNTF has good compatibility with most low-melting-point explosives, and the peak temperature change of the composite system formed by melting casting is not significant. Among them, DNTF has the best compatibility with MTNP, TNT, DNAN, and DFTNAN, moderate compatibility with DNP, and the worst compatibility with DNMT.(2)Comparing DNTF with several low-melting-point-explosive systems, it can be found that the combination of DNTF and TNT has the most significant effect on reducing mechanical sensitivity. The friction sensitivity decreased by 76%, the drop height increased by 3.02 cm, and the impact sensitivity decreased. Due to the high friction sensitivity of DNTF, it is prone to friction ignition and explosion, resulting in detonation. The addition of TNT significantly reduces the friction sensitivity of DNTF, helping to improve the safety of DNTF explosives.(3)The simulation results of binding energy are almost in agreement with the compatibility test, indicating that many explosives with good compatibility can be screened by calculating the binding energy. DFTNAN and MTNP are more stable than DNTF and have stronger intermolecular forces when combined with DNTF. The intermolecular force generated by mixing other low-melting-point explosives with DNTF is lesser than that of DNTF. The MTNP/DNTF composite system has the highest cohesive energy density and requires the highest amount of energy to undergo phase transition when heated compared to other mixed systems, making it the least sensitive to heat. However, the DNTF/DNMT system has the lowest cohesive energy density and is the most sensitive to heat.

## Figures and Tables

**Figure 1 molecules-29-03757-f001:**
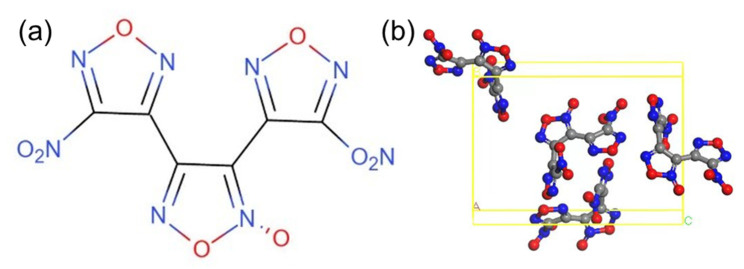
The structures of DNTF: (**a**) chemical structure; (**b**) crystal structure.

**Figure 2 molecules-29-03757-f002:**
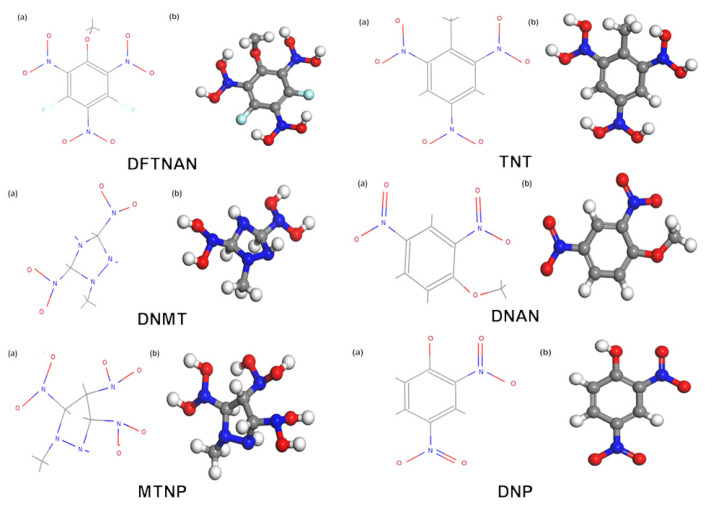
Molecular structural formulas (**a**) and configurations (**b**) of six low-melting-point explosives.

**Figure 3 molecules-29-03757-f003:**
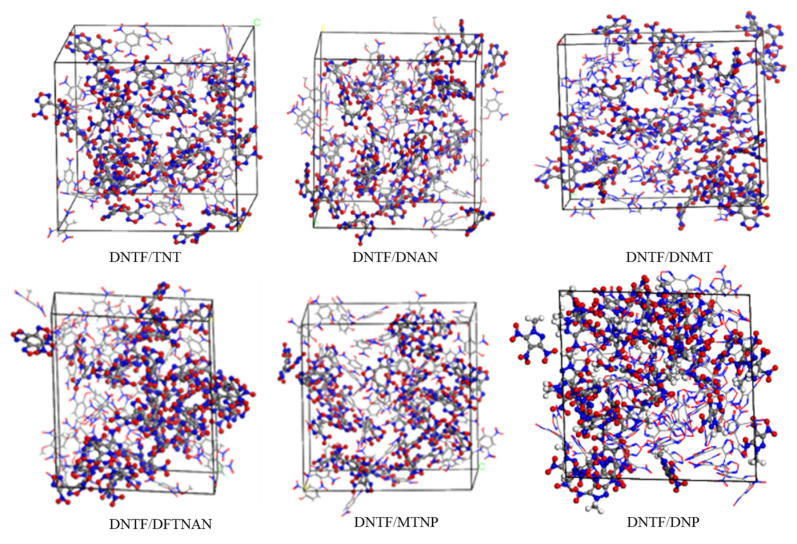
Initial models of DNTF and low-melting-point explosives.

**Figure 4 molecules-29-03757-f004:**
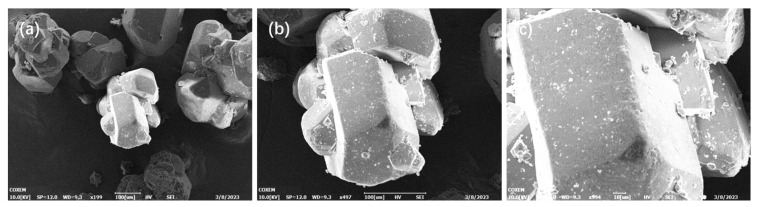
DNTF morphology: (**a**) 200 times; (**b**) 500 times; (**c**) 1000 times.

**Figure 5 molecules-29-03757-f005:**
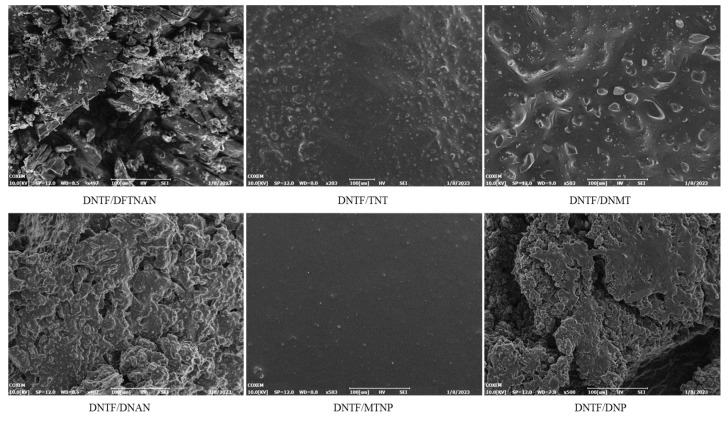
Morphology of the mixtures with DNTF and low-melting-point explosives (500 times).

**Figure 6 molecules-29-03757-f006:**
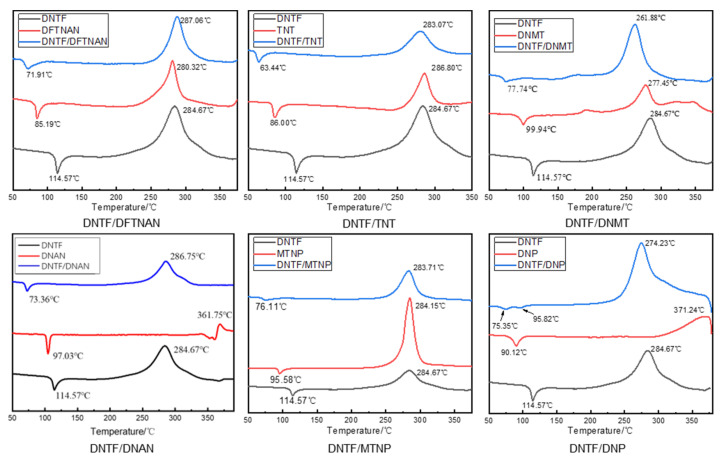
DSC curve of the mixtures with DNTF and low-melting-point explosives.

**Table 1 molecules-29-03757-t001:** Evaluation standards of the compatibility for explosives and contact materials.

Criteria ΔTp (°C)	Rating	Note
Less than or equal to 2	A, compatible or good compatibility	Safe for use in any explosive design
3~5	B, slightly sensitized or fair compatibility	Safe for use in testing or when the device will be used for a very short period of time; not to be used as a binder material or when long-term storage is desired
6~15	C, sensitized or poor compatibility	Not recommended for use with explosive items
Above 15	D, hazardous or bad compatibility	Hazardous. Do not use under any conditions

**Table 2 molecules-29-03757-t002:** Initial parameters of calculation model of mixed system.

Sample		Molecular Number of Low-Melting-Point Explosive	Molecular Number of DNTF	Atom Number	**Density** **(g·cm^−3^)**
Ⅰ	DNTF	30	30	1320	1.853
Ⅱ	DNTF/DFTNAN	34	30	1386	1.821
Ⅲ	DNTF/TNT	42	30	1542	1.689
Ⅳ	DNTF/DNMT	54	30	1470	1.727
Ⅴ	DNTF/DNAN	48	30	1600	1.595
Ⅵ	DNTF/MTNP	43	30	1434	1.803
Ⅶ	DNTF/DNP	47	30	1510	1.687

**Table 3 molecules-29-03757-t003:** Decomposition temperatures of binary systems obtained by DSC.

Binary System	Single System	Peak Temperature			
		Tp1 (°C)	Tp2 (°C)	ΔTp (°C)	Rating
DNTF/DFTNAN	DNTF	284.67	287.06	2.39	B
DNTF/TNT	DNTF	284.67	283.07	1.60	A
DNTF/DNMT	DNTF	284.67	261.88	22.79	D
DNTF/DNAN	DNTF	284.67	286.75	2.08	B
DNTF/MTNP	DNTF	284.67	283.71	0.96	A
DNTF/DNP	DNTF	284.67	274.23	10.44	C

Note: Binary system, DNTF/low-melting-point-explosive binary system; single system, DNTF single-energetic-material system; $T_{p1}$, maximum exothermic peak temperature of a single system; maximum exothermic peak temperature of mixture system; $∆T_{p}=\left|T_{p1}-T_{p2}\right|$.

**Table 4 molecules-29-03757-t004:** Summary of impact sensitivity and friction sensitivity results.

Sample (Ratio)	Friction Sensitivity	Impact Sensitivity
	P/%	H_50_/cm
DNTF	100%	43.65
DFTNAN	0%	40.97
TNT	15%	≥100
DNMT	60%	92.7
DNAN	16%	≥100
MTNP	0%	≥100
DNP	32%	≥100
DNTF/DFTNAN	60%	39.80
DNTF/TNT	24%	46.67
DNTF/DNMT	52%	38.90
DNTF/DNAN	36%	47.86
DNTF/MTNP	40%	40.56
DNTF/DNP	44%	35.48

**Table 5 molecules-29-03757-t005:** Corrected binding energy of the substituted models with different molecular molar ratios.

System	E_total_/kJ·mol^−1^	E_DNTF_/kJ·mol^−1^	E_layer2_/kJ·mol^−1^	E_bind_/kJ·mol^−1^
DNTF/DFTNAN	−3994.21	−686.26	−2212.49	1095.46
DNTF/TNT	−1321.95	2016.10	−2150.20	1187.85
DNTF/DNMT	−9630.89	−6611.40	−2186.12	813.37
DNTF/DNAN	−741.00	2511.64	−2080.46	1172.18
DNTF/MTNP	−7354.68	−4039.98	−2126.76	1187.94
DNTF/DNP	−1322.16	1886.79	−2202.72	1016.23

Note: E_DNTF_ and E_LMPE_ are the residual energy of DNTF and low-melting-point explosives in the mixed system, respectively. E_bind_ is the interaction energy between the two components. E_total_ is the total interaction energy.

**Table 6 molecules-29-03757-t006:** Corrected binding energy Eb* of the substituted models with different molecular molar ratios (300 K).

System	E_total_/kJ·mol^−1^	E_DNTF_/kJ·mol^−1^	E_layer2_/kJ·mol^−1^	E_bind_/kJ·mol^−1^
DNTF/DFTNAN	−3994.21	−686.26	−2212.49	1095.46
DNTF/TNT	−1321.95	2016.10	−2150.20	1187.85
DNTF/DNMT	−9640.82	−6365.79	−2461.84	813.19
DNTF/DNAN	−741.00	2511.64	−2080.46	1172.18
DNTF/DNP	−1332.17	1886.80	−2202.72	1016.25

**Table 7 molecules-29-03757-t007:** Corrected binding energy Eb* of the substituted models with different molecular molar ratios (2000 K).

System	E_total_/kJ·mol^−1 ^	E_DNTF_/kJ·mol^−1 ^	E_LMPE_/kJ·mol^−1 ^	E_bind_/kJ·mol^−1 ^
DNTF/DFTNAN	4656.87	3433.6	1293.33	70.06
DNTF/TNT	7988.16	6686.18	1389.42	87.44
DNTF/DNMT	−1002.00	−2225.50	1300.39	76.89
DNTF/DNAN	8653.11	7494.62	1279.60	121.11
DNTF/DNP	7733.31	6341.74	1454.05	62.48

**Table 8 molecules-29-03757-t008:** Corrected binding energy Eb* of the substituted models with different molecular molar ratios (3000 K).

System	E_total_/kJ·mol^−1^	E_DNTF_/kJ·mol^−1^	E_LMPE_/kJ·mol^−1^	E_bind_/kJ·mol^−1^
DNTF/DFTNAN	−1592.53	−1386.14	799.27	1005.66
DNTF/TNT	1442.12	1356.91	878.81	793.6
DNTF/DNMT	−4211.64	−4087.31	784.02	908.35
DNTF/DNAN	1479.18	1653.01	884.42	1058.25

**Table 9 molecules-29-03757-t009:** Cohesive energy density of the substituted models with molecular molar ratios.

Model	CED (Kcal/cm^3^)	Evdw (Kcal/cm^3^)	Eelectrostatic (Kcal/cm^3^)
DNTF/MTNP	10.894	3.7936	6.9849
DNTF/DNAN	7.8535	3.7790	3.9722
DNTF/TNT	8.9443	3.5996	5.2394
DNTF/DNP	8.6095	4.0168	4.6480
DNTF/DNMT	7.5919	3.8999	3.6920
DNTF/DFTNAN	9.7717	3.6557	6.0040

Note: CED = E_vdw_ + E_electrostatic_.

## Data Availability

Data are contained within the article.

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
