# Peer review of "Experiment and Molecular Dynamic Simulation on Interactions between 3,4-Bis(3-nitrofurazan-4-yl) Furoxan (DNTF) and Some Low-Melting-Point Explosives"

_molecules, 2024, doi:10.3390/molecules29163757_

Round 1

Reviewer 1 Report

Comments and Suggestions for Authors

In this submission to Molecules, the authors explore the effect of DNTF on low melting point molten carrier explosives using differential scanning calorimetry and mechanical sensitivity. The authors calculated the compatibility and cohesive energy density between DNTF and other low melting point explosives using Materials Studio. Further comments on their simulations are given below. The authors' results showed that DNTF is compatible with the most low melting point explosives. The authors find that DNTF has the best compatibility with MTNP, TNT and DNAN. The authors' sensitivity test results indicate that the combination of DNTF and TNT has the most significant reduction in mechanical sensitivity. The authors find that MTNP/DNTF system requires the most energy to phase change when heated compared to other mixed systems and the DNMT/DNMT system has the lowest cohesive energy density and is the most sensitive to heat.

I find this manuscript to be of interest to computational (and experimental) chemists as well as readers of this journal. As such, I am generally supportive of publication with a few minor notes. Specifically, the authors use classical molecular dynamics to explore interactions in molecular explosives. However, there has been prior work on using first-principles-based molecular dynamics with DFT to explore energetic materials, which should be noted as an alternative approach: J. Phys. Chem. B 2008, 112, 11005–11013 and J. Phys. Chem. C 2021, 125, 21922–21932. In particular, these prior studies used ab initio molecular dynamics to explore these energetic systems. I am not asking the authors to do ab initio molecular dynamics calculations, but this alternative approach for calculating interactions in energetic materials should also be noted. With this minor edit, I would be willing to re-review this manuscript for publication in Molecules.

Reviewer 2 Report

Comments and Suggestions for Authors

The article, titled "Experiment and Molecular Dynamic Simulation on Interactions between 3,4-bis(3-nitrofurazan-4-yl) furoxan (DNTF) with some Low Melting Point Explosives" describes the results of preparing binary mixtures of high-energy compounds to improve properties such as friction sensitivity and impact sensitivity. The article yields interesting data, but some adjustments are needed before it can be published in Molecules:

1. It is necessary to add a reference to the literature on where the crystal structure in Figure 1b is given.

2. A summarized figure with structural formulas of all explosive compounds used in the article should be given.

3. On page 2, a definition of all abbreviations used in the text should be given: 5-ATEZN, NTO, TATB, NC, GAP, etc.

4. It remains unclear where the criteria given in Table 1 come from. An appropriate reference should be provided.

5. On page 5, it is necessary to clarify the resolution with which the photographs in Figure 4 were taken.

Author Response

Please refer to the attachment for the response comments.
